# Cu-Doped Hollow Bioactive Glass Nanoparticles for Bone Infection Treatment

**DOI:** 10.3390/pharmaceutics14040845

**Published:** 2022-04-12

**Authors:** Javier Jiménez-Holguín, Sandra Sánchez-Salcedo, Mónica Cicuéndez, María Vallet-Regí, Antonio J. Salinas

**Affiliations:** 1Departamento de Química en Ciencias Farmacéuticas, Facultad de Farmacia, Universidad Complutense de Madrid, UCM, Instituto de Investigación Hospital 12 de Octubre, Imas12, 28040 Madrid, Spain; javiej03@ucm.es (J.J.-H.); mcicuendez@farm.ucm.es (M.C.); vallet@ucm.es (M.V.-R.); 2Networking Research Center on Bioengineering, Biomaterials and Nanomedicine, CIBER-BBN, 28040 Madrid, Spain

**Keywords:** hollow nanoparticles, mesoporous glasses, copper, antibacterial, infection, pre-osteoblasts

## Abstract

In search of new approaches to treat bone infection and prevent drug resistance development, a nanosystem based on hollow bioactive glass nanoparticles (HBGN) of composition 79.5SiO_2_-(18-x)CaO-2.5P_2_O_5_-xCuO (x = 0, 2.5 or 5 mol-% CuO) was developed. The objective of the study was to evaluate the capacity of the HBGN to be used as a nanocarrier of the broad-spectrum antibiotic danofloxacin and source of bactericidal Cu^2+^ ions. Core-shell nanoparticles with specific surface areas close to 800 m^2^/g and pore volumes around 1 cm^3^/g were obtained by using hexadecyltrimethylammonium bromide (CTAB) and poly(styrene)-block-poly(acrylic acid) (PS-b-PAA) as structure-directing agents. Flow cytometry studies showed the cytocompatibility of the nanoparticles in MC3T3-E1 pre-osteoblastic cell cultures. Ion release studies confirmed the release of non-cytotoxic concentrations of Cu^2+^ ions within the therapeutic range. Moreover, it was shown that the inclusion of copper in the system resulted in a more gradual release of danofloxacin that was extended over one week. The bactericidal activity of the nanosystem was evaluated with *E. coli* and *S. aureus* strains. Nanoparticles with copper were not able to reduce bacterial viability by themselves and Cu-free HBGN failed to reduce bacterial growth, despite releasing higher antibiotic concentrations. However, HBGN enriched with copper and danofloxacin drastically reduced bacterial growth in sessile, planktonic and biofilm states, which was attributed to a synergistic effect between the action of Cu^2+^ ions and danofloxacin. Therefore, the nanosystem here investigated is a promising candidate as an alternative for the local treatment of bone infections.

## 1. Introduction

The self-healing capacity of the skeletal system is impaired in certain clinical situa-tions, for instance, in bone defects greater than 50% of the diameter of bone (1–2 cm) [1] or when there is not an adequate osteoblast/osteoclast balance [2]. The situation worsens when the bone tissue is exposed to the environment as the result of an open wound. In those cases, bacterial infections may occur, caused in most cases by Gram-positive cocci. Indeed, more than 60% of cases are due to the Staphylococci genus [3]. Once bacterial colonization in bone occurs there are three possible outcomes: the resolution of the infection, the conversion of the bacteria to a quiescent phenotype, or the formation of a chronic infection [4]. In the latter case, the usual procedure consists of treatment with several bactericidal drugs for a prolonged period of time either parenterally or orally, reaching very high systemic concentrations to achieve the minimum inhibitory concentration in bone [5,6,7,8]. In 20% of cases, treatment fails [9], due to: (i) the formation of a dense layer of exopolysaccharides generated by bacteria, known as biofilm, which provides protection against the effect of antibiotics [10,11], (ii) the formation of small variant colonies (SCVs) with low metabolism that elude the effect of antibiotics [12,13,14] and (iii) the emergence of antibiotic-resistant strains as a consequence of the selective pressure in the natural environment [15]. Due to the difficult access to the infected tissue inside, surgeries are often used in which the necrotic tissue must be removed and, when the bone defect that occurs is large, a bioactive material must be applied to support and promote healing, such as a scaffold or a bioactive glass-based paste [16]. In severe cases, is necessary to remove the limb to prevent the infection from spreading [17].

Bioactive glasses, discovered in 1971, are characterized by binding strongly to bone without the fibrous capsule formed being detected when non-bioactive materials are implanted. Indeed, such capsules cause the failure of many implants [18]. Later on, bioactive glasses with improved properties for bone regeneration were obtained by using wet synthesis methods, such as sol–gel processing. When surfactants were included to act as templates, the so-called mesoporous bioactive glasses (MBG) were obtained, most of them in the SiO_2_-CaO-P_2_O_5_ system [19]. One of the latest advances in this field includes the development of hollow bioactive glass nanoparticles (HBGN) that exhibit additional capabilities compared with MBG including parenteral administration, increased textural properties, low toxicity and easy excretion [20]. These nanoparticles have a structure composed of the shell and a core hat that is achieved by employing two structure-directing agents: hexadecyltrimethylammonium bromide (CTAB) used to obtain the radial mesoporous order and poly(styrene)-block-poly(acrylic acid) (PS-b-PAA) to obtain the core and the internal porosity [21]. The HBGN’s structure results in a high specific surface area and pore volume that enables it to transport biomolecules and release them in a sustained manner in aqueous media [22,23,24]. Bioactive SiO_2_-CaO-P_2_O_5_ glasses have a reactive surface that, when in contact with body fluids, leads to nucleation and formation of a hydroxycarbonate apatite (HCA) phase [25,26] similar to the mineral component of bone, thus favoring its binding to bone. The release of ions from these glasses promotes the proliferation and differentiation of the cells involved in osteogenesis [25]. The surface of these materials has plenty of silanol groups that allow functionalization with organosilanes accompanied by different functional groups (amine, carboxyl, thiol, etc.) and, thus, modify their surface properties and increase their therapeutical applications [27,28].

Another characteristic of the glasses is the possibility of incorporating in their structure metals with therapeutic capabilities, such as silver [29], zinc [30], cobalt [31] or copper [32], among others. It is well known in the literature the ability of copper to induce angiogenesis by stimulating the production of fibroblast growth factor-2 (FGF-2) [33] and vascular endothelial growth factor (VEGF) [34,35]. Moreover, copper ions have the ability to increase the differentiation of mesenchymal cells towards osteogenic lines [36] and to inhibit osteoclast activity [37]. Copper ions also participate in the crosslinking of the collagen fibers necessary for bone ossification [38,39]. In addition, the bactericidal activity of copper against *E. coli*, *B. subtilis* and *S. aureus* has been described in the literature [40,41]. Although the mechanisms of action are still under discussion, it has been reported that Cu^2+^ ions bind to the bacterial cell wall by electrostatic attraction, due to the fact that the bacterial wall is densely populated by negatively charged peptidoglycan molecules, the attraction being greater for Gram-negative bacteria [42]. This leads to a solidification of the structural proteins that compose it, irreversibly damaging it [43,44]. The damage may be mediated by the production of reactive hydroxyl radicals that are known to be involved in numerous reactions detrimental to proper cell development, such as oxidation of proteins and lipids [45].

The present work is focused on the development of HBGN within the SiO_2_-CaO-P_2_O_5_ system enriched with CuO (0, 2.5 or 5 mol-%). Their cytocompatibility with MC3T3-E1 osteoprogenitor cells will be studied. Moreover, the antibacterial efficacy of the three nanosystems before and after loading with danofloxacin will be evaluated. Danofloxacin is an antibiotic of the family of fluoroquinolones actives against most of the Gram-negative bacteria such as *Escherichia coli*, *Klebsiella*, and *Salmonella*, as well as some Gram-positive ones, including *Staphylococcus aureus* and *Streptococcus pneumoniae* [46]. Danofloxacin was selected due to its small size, broad spectrum effect and low minimal inhibitory concentration (MIC) [47]. Finally, the antibacterial efficacy of our nanosystem loaded with danofloxacin against *E. coli* and *S. aureus*, as the Gram-negative and -positive bacteria models, respectively, in sessile and planktonic states will be investigated. The global aim of this study is the search of alternative therapies for the treatment of acute and chronic bone infections.

## 2. Materials and Methods

### 2.1. Synthesis of the Nanoparticles

Three HBGN compositions were synthesized with increasing amounts of copper within the 79.5SiO_2_-(18-x) CaO-2.5P_2_O_5_-xCuO (x = 0, 2.5, 5 mol-% CuO) system using the protocol described by Li et al. [21]. This methodology consists of preparing an emulsion of an organic/aqueous solution in which two structure directing agents are dissolved: CTAB and PS-b-PAA of an average molecular weight, M_n_ 38,000 Da, 20 wt-% PAA. On the one hand, an inorganic solution (IS) was prepared by dissolving CTAB and concentrated NH_3_ in distilled water, while in another container an organic solution (OS) was prepared by dissolving PS-b-PAA in tetrahydrofuran (THF), keeping both mixtures under stirring at 37 °C for 30 min to later pour, abruptly, the OS over the IS, leaving it under stirring for 20 min. Subsequently, the appropriate amounts of triethyl phosphate (TEP), Ca(NO_3_)_2_·4H_2_O, Cu(NO_3_)_2_·2.5H_2_O and tetraethyl orthosilicate (TEOS), previously dissolved in ethanol, for TEP and TEOS, or water, for Ca(NO_3_)_2_·4H_2_O, were added sequentially and spaced in time by 20 min. The mixture was kept stirring for 24 h at 37 °C and covered with Parafilm to avoid evaporation. The resulting product was centrifuged at 12,000 rpm for 12 min at 4 °C and washed once with H_2_O and twice with H_2_O/ethanol (50%). After 24 h of drying at 30 °C, the samples were calcined 4 h at 550 °C using a ramp of 1 °C/m. The steps carried out during the synthesis process are shown in Figure 1 and the amounts of reactants used are shown in Table 1. All reagents were purchased from Sigma-Aldrich, St. Louis, MO, USA).

### 2.2. Physicochemical Characterization

To evaluate the textural characteristics of the materials and verify their suitability for this study, they were characterized by nitrogen adsorption-desorption using a Micromeritics 3 Flex (Norcross, GA, USA) that allows the specific surface, S_BET_, to be obtained by the Brunauer–Emmett–Teller method (BET) [48] and pore size distribution using the Barret–Joyner–Halenda method (BJH) [49]. The structure of the nanoparticles could be analyzed using transmission electron microscope (TEM) micrographs in a JEM-1400 JEOL operating between 40 and 120 kV (Tokyo, Japan). Energy dispersive X-ray fluorescence (EDS) was carried out with a CCD camera (KeenVIEW) operating at 120 kV in combination with TEM to determine the atomic composition of the samples. The intensity of the static electric field of the nanoparticles was determined using a Zetasizer Nano-ZS dynamic light scattering device (DLS) (Malvern Instruments, Malvern, UK).

### 2.3. Ion Release Assay

Ion release assay was performed in α-MEM at 37 °C. A 12 transwell plate with two different compartments (sample and analysis) was employed for the experiment. Both compartments are separated by a membrane (12 kDa molecular weight) that only allows the diffusion of antibiotic and medium. A sample of 4.15 mg of each HBGN composition was suspended in 0.5 mL of α-MEM and was placed in the sample compartment, and 2 mL of fresh α-MEM was placed in the analysis compartment. An aliquot of α-MEM located in the analysis compartment was extracted and refilled at 2, 24, 48, 72 and 144 h. The cumulative concentrations of Ca^2+^, and Cu^2+^ in the culture medium were obtained by inductively coupled plasma/optical spectrometry (ICP/OES) using an OPTIMA 3300 DV device (Perkin Elmer, Waltham, MA, USA). Three measurements were made on each of the two replicates per composition examined.

### 2.4. Loading and Release of Danofloxacin

The three HBGN compositions were loaded with danofloxacin (5 mg of HBN and 0.6 mg of danofloxacin in 10 mL of pure ethanol) for 24 h at 4 °C under stirring and darkness. The resulting product was centrifuged at 10,000 rpm at 4 °C for 10 min. After this, it was washed twice with 5 mL of pure ethanol, allowed to dry at 30 °C for 24 h. The amount of danofloxacin loaded was determined by thermogravimetric studies and differential thermal analysis (TG/DTA) in a dynamic air atmosphere between 110 °C and 600 °C with a ramp of 5 °C/min, air flow: 100 mL/min) in a Perkin Elmer Pyris Diamond system (Waltham, MA, USA).

The danofloxacin release was carried out in phosphate-buffered saline (PBS) medium (Gibco, Thermo Fisher Scientific, Wilmington, DE, USA) at 37 °C and physiological pH of 7.4. A 12 transwell plate was employed for the experiment. A measure of 0.5 mL of a suspension of HBGN loaded in PBS (2 mg/mL) was placed in the sample compartment and 1.5 mL of fresh PBS in the analysis compartment. PBS placed in the analysis compartment were collected and refilled every 30 min the first day until 6 h, then every hour the second day, every 2 h the third day and go on, up to 6 d. Collected samples were measured by fluorescence spectroscopy using a Biotek Powerwave XS fluorimeter (version 1.00.14 of the Gen5 program). For danofloxacin, λ_ex_ = 280 nm and λ_em_ = 430 nm were used in the fluorimeter. Three measurements were made on each of the two replicates per composition examined.

### 2.5. In Vitro Assays in Pre-Osteoblastic Mammalian Cells

All the tests were carried out under sterility, for this, the nanoparticles were irradiated with ultraviolet light for 20 min. Pre-osteoblastic cells from MC3T3-E1 mice (subclone 4, CRL-2593; ETCC, Manassas, VA, USA) were used for these tests. Cells were cultured in a supplemented culture medium, α-MEM with 10% fetal bovine serum (FBS, Gibco, Thermo Fisher Scientific, Wilmington, DE, USA), 1% penicillin-streptomycin and 5 mM L-glutamine (Gibco, Thermo Fisher Scientific, Wilmington, DE, USA) with a saturation of 5% CO_2_ at 37 °C. Cells were washed with PBS, pH 7.4, and harvested with 5 mL of 0.25% trypsin-EDTA (Gibco, Thermo Fisher Scientific, Wilmington, DE, USA). The cell suspension was centrifuged for 10 min at 1200 rpm. The resulting cells were resuspended with fresh supplemented media.

#### 2.5.1. Cytotoxicity Assays in Pre-Osteoblastic Mammalian Cells

For this test, three replicates were arranged for the control and three for each material. The 1 × 10^4^ cells/mL MC3T3-E1 were exposed to two concentrations of nanoparticles (200 µg/mL and 300 µg/mL) for 2 h at 37 °C and in an atmosphere with 5% CO_2_. After this period, the cells were washed with PBS and were incubated again with fresh supplemented medium for 24 and 72 h. The cytotoxicity of the material was evaluated by means of cell proliferation, for this the MTS methodology (CellTiter 96 AQueous Assay (Promega, Madison, WI, USA) based on the reduction of compound 3[4,5-dimethylthiazole- 2-yl]-5-(3-carboxymethoxyphenyl)-2-(4-sulfophenyl)-2H-tetrazolium (MTS) by mitochondrial NADH using an intermediate electron acceptor, phenazine ethyl sulphate (PES) [50].

For this purpose, the culture medium was removed, the cells were washed with PBS, 100 µL of supplemented culture medium, 20 µL of MTS solution were added and the cells were incubated for 3 h at 37 °C and 5% CO_2_. After this incubation, the resulting product was collected for analysis on the Helios Zeta UV–Vis spectrophotometer at a wavelength of 490 nm.

#### 2.5.2. Internalization of HBGN and Morphology Studies

To observe the location of the HBGN inside the cells, HBGN labelled with fluorescein-5-isothiocyanate (FTIC) were synthesized. For this, 2.2 µL of (3-aminopropyl) triethoxysilane (APTS) was mixed with 2 mg of FTIC previously dissolved in 200 µL of ethanol and incubated under shaking and darkness for 2 h. After this incubation, 2.4 µL of this resulting solution was mixed with 1.38 µL of ethanol and 455 µL of TEOS. This mixture was added during the HBGN synthesis described in Section 2.1, performing the remaining synthesis process in the dark. To remove the surfactant, 50 mg of HBGN labelled with FTIC were mixed with 20 mL of extractor solution (20 g of NH_4_NO_3_ dissolved in 1 L of ethanol) and 20 mL of ethanol, under reflux and dark at 85 °C for 4 h. The resulting product was centrifuged at 12,000 rpm at 15 °C for 10 min, to later carry out one wash with ethanol, two with THF and finally, one wash with ethanol. The resulting material was left to dry at 60 °C for 24 h for its later use.

HBGN samples labelled with FTIC were put in contact with 5 × 10^4^ cells of the MC3T3-E1 line at a concentration of 200 µg/mL for 2 h at 37 °C and 5% CO_2_. After this time, the cells were washed with PBS and incubated 24 h at 37 °C and 5% CO_2_. The next step consisted of fixing with ethanol (75% purity) for 2 min at 37 °C, then Phalloidin-ATTO 565 (1:40 dilution, Molecular Probes) was added for 10 min in the dark. The medium was re-removed for Fluoroshield staining with 2- (4-amidinophenyl)-6-indolecarbamidine dihydrochloride (DAPI) (1:1000, Sigma Aldrich, St. Louis, MO, USA) for 10 min in the dark. Finally, each well was washed twice and kept in PBS until microscopic analysis. The images were obtained with an EVOS FL Cell Imaging System inverted optical microscope equipped with three types of led light (IEX (nm); IEM (nm)): DAPI (357/44; 447/60), GFP (470/22; 525/50), RFP (531/40; 593/40) from AMG (Advance Microscopy Group, Bothell, WA, USA). The red channel was used to observe the cytoskeleton, blue, to observe the cell nucleus and the green to see HBGN.

#### 2.5.3. Flow Cytometry Assays

For this tests, 2.5 × 10^4^ MC3T3-E1 cells were seeded with 2 mL of supplemented medium per well. Three replicas for sample were employed, whereas four replicas of control cells were used, one for each assay condition. At 24 h after the seeding, cells were washed with PBS and 50 μg/mL of each HBGN composition, dispersed in supplemented medium, were added for 3 d. A total of 10^5^ cells were analyzed in each sample in order to ensure a correct statistical significance. Analysis were carried out in a FACScalibur flow cytometer (Becton Dickinson, San Jose, CA, USA).

#### 2.5.4. Cell Size and Complexity Analysis

Cell size and complexity in presence of each HBGN composition were researched. For that, forward-scatter angle (FSC) and side-scatter angle (SSC) were analyzed.

#### 2.5.5. Cell Viability, Intracellular Reactive Oxygen Species (ROS) Content

ROS intracellular content was determined by addition of 100 µM 2′,7′-dichlorofluorescein diacetate (DCFH/DA) at 37 °C for 40 min. Cell viability was evaluated by adding 0.005% (wt/vol) propidium iodide (PI) (Sigma-Aldrich, St. Louis, MO, USA) in PBS into the samples to stain the dead cells. DCF fluorescence was excited at 488 nm and measured with a 530/30 nm band pass filter, meanwhile PI fluorescence was excited at 488 nm and the emission was measured with a 670 nm LP.

### 2.6. In Vitro Microbiology Assays

For these studies, Gram-negative Escherichia coli (ATCC25922) and Gram-positive *Staphylococcus aureus* (ATCC29213) were used as bacteria models. To work with them, lysogeny broth media (LB) and Todd Hewitt Broth medium (THB) were used, as well as LB agar plates and tryptone-soy agar (TSA), respectively. With the help of a seeding loop, a colony of each species was inoculated into 20 mL of their respective culture medium to incubate them for 24 h at 37 °C while shaking. The resulting suspension was diluted 1/10 in fresh medium and incubated again at 37 °C until obtaining an optical density (OD) of 0.4 at 600 nm, thus estimating a concentration of 2 × 10^8^ colony forming units (CFUs)/mL with which to perform the rest of the tests.

#### 2.6.1. Bacterial Growth Inhibition Tests in the Planktonic State

The HBGN without loading and loaded with danofloxacin, were diluted to different concentrations in 96-well plates, arranging 3 replicates per material in such a way that 200, 100 and 10 µg/mL were tested: and 260, 130 and 20 µg/mL for *E. coli* and *S. aureus,* respectively, with a final concentration of 7.1 × 10^5^ CFUs/mL for both bacterial strains. The bacteria were kept in culture at 37 °C and under shaking conditions while aliquots were extracted at 24 h. These aliquots were serially diluted until obtaining 10^−10^ dilutions in the case of controls and unloaded samples; and up to 10^−4^ in the samples loaded with danofloxacin, to finally sow 4 drops of 10 µL per dilution on agar plates that were kept in static incubation at 37 °C for 24 h. Once this period had passed, the number of CFUs were counted and the dilution factor was applied. Minimum bactericidal concentration (MBC) was determined by the lowest concentration of antibacterial agent that reduces the viability of the initial bacterial inoculum by 99.9%.

#### 2.6.2. Halo Inhibition Assays

The unloaded and danofloxacin-loaded HBGN were introduced into PBS at concentrations of 300, 200 and 100 µg/mL in a 12-well plate, using 3 replicates per control and samples. At 6 and 24 h of incubation at 37 °C, 50 µL aliquots were loaded with sponges, in the form of 9 mm discs, which were placed on agar plates pre-seeded with the bacterial strains *S. aureus* or *E. coli*. After addition, these plates were statically incubated at 37 °C for 24 h. Subsequently, the diameters of the bacterial growth inhibition halos from the sponge used were analyzed.

#### 2.6.3. Biofilm Degradation Assay

Measures of 2 × 10^8^ CFUs/mL of *S. aureus* were seeded in 12-well plates with a circular glass slide and cultured with THB enriched with 4% sucrose at 37 °C under shaking (100 rpm) for 24 h. Afterwards, this medium was replaced with fresh medium without sucrose and it was kept for another 24 h until the biofilm was obtained. The slides with the biofilm were transferred to 12-well plates with transwell and 200 µg/mL of the three danofloxacin-loaded HBGN compositions. After 24 h of incubation with shaking at 37 °C, the biofilms were extracted and stained using the BacLight bacterial viability kit (Thermo Fisher Scientific, Invitrogen, Waltham, MA, USA). Images were obtained by means of an OLYMPUS FV1200 laser confocal microscope (OLYMPUS, Tokyo, Japan), using 60× FLUOR immersion lenses in water (NA = 1.0). The micrographs were obtained using the 3D Imaris software that was used to convert each of the 2D images into 3D micrographs, using the Z sections and an algorithm that shows the maximum pixel value of each 1 mm Z cut. Live bacteria were observed in green (SYTO 9), dead ones in red (PI) and the biofilm matrix in blue (Calcofluor).

### 2.7. Statistical Analysis

The results were expressed as a measure of mean standard deviation (SEM). Statistical analysis was performed with the non-parametric Kruskal–Wallis test and Dunn’s post hoc test. A value of *p* < 0.05 was considered significant.

## 3. Results and Discussion

### 3.1. Characterization of the Microstructure of the Nanoparticles

The thermogravimetric analysis confirmed the complete elimination of both surfactants (CTAB and PS-b-PAA) from the three HBGN compositions (results not shown), which during the article will be named as 0%, 2.5% and 5% Cu-HBGN.

The nanoparticles were physicochemically characterized. TEM analysis allowed us to verify their nanostructure and EDS to find out if the copper ions were incorporated into the glass structure. The TEM micrographs depicted in Figure 1 show non-aggregated hollow nanospheres with a diameter of 190 ± 65 nm. Table 2 includes the values for each case showing that no significant differences were observed between the sizes obtained from the different compositions. As can be seen in the figure, HBGN are compartmentalized in what is known as “core” and “shell” nanostructures. The images show that the core occupies between 60–70% of the size of the nanoparticles. Moreover, radial pores can be observed in the shell, the formation of which was attributed to the use of CTAB in their syntheses whereas the core was formed by the presence of the second surfactant, that is, PS-b-PAA.

Furthermore, EDS analysis allowed us to determine the atomic percentage of copper incorporated into the HBGN structure. The results are gathered in Table 2, obtaining values very close to the nominal calculated. It should be noted that the incorporation of phosphorus was so low that it is below the detection limits of the technique. Phosphorus integration is usually reduced due to the slow hydrolysis rate of the phosphorus precursor TEP [51]. Despite the low incorporation rate also observed in other works [52,53], it was decided to keep the addition of TEP as it allows a higher incorporation of calcium into the nanoparticles [53].

Figure 2A shows the isotherms obtained from the N_2_ adsorption. The three HBGN compositions are characterized by an adsorption isotherm type IV characteristic of mesoporous materials, and irregularly shaped hysteresis cycles revealing the presence of at least two types of pores. These cycles can be classified as H2 types, characteristic of inkwell-shaped pores [54]. Probably, the pores located inside the nucleus are connected with the radial pores located in the “shell”, as previously indicated in the bibliography [21,55]. All three HBGN compositions exhibited a specific surface area of more than 700 m^2^ g^−1^ (Table 2). The highest specific surface area was obtained for the Cu-free composition 825 m^2^ g^−1^, as opposed to the Cu-enriched compositions that present 731 and 739 m^2^ g^−1^. These results show that HBGN-Cu shows superior textural properties to the other nanoparticles [56,57,58], nanoparticles doped with other therapeutic ions [59,60,61,62], and even other hollow nanoparticles [63,64,65]. The same occurred with the pore volume, which decreased from 1.22 to 0.75 cm^3^ g^−1^. The slight decrease in textural properties is an expected result, considering previous work in the literature in which metal ions were included as modifiers in the silica network [30,31,66,67]. Furthermore, pore size, increased slightly. However, the textural properties obtained between the Cu-doped and undoped compositions were similar, and their difference does not represent a significant loss in their potential application. Figure 2B shows the presence of three maxima: a maximum of about 1 nm in the three compositions that can be attributed to a type of pore produced by the interaction between the two surfactants; the maximum around 2.6 nm corresponding to the pore produced by CTAB; and a third maximum that varies depending on the composition due to the inner cavity called “core”. This last cavity was larger in the Cu-free composition.

The electrical environment of the HBGN, was studied by ζ potential. The three compositions revealed a negative charge of between −28.4 and −17.3 mV, a characteristic that has been shown to increase proliferation and the rate of bone formation [68,69]. Surface silanol groups are responsible for said negative charge, a typical characteristic of bioglasses, which allows for the possibility of functionalizing the said material to increase its therapeutic applications [70]. 

To investigate the adsorption capacity of the HBGN and their use as a nanocarrier, HBGN were loaded with danofloxacin. The amount loaded in the HBGN was determined by TG/DTA. Table 2 shows the amount of danofloxacin loaded inside the HBGN. As can be seen, the adsorption of danofloxacin is similar in the three compositions, being slightly higher in those doped with Cu. In the literature, it is described that the incorporation of Cu into the MBG increases the loading capacity of certain drugs [71]. In our case, this may be due to the interaction between the Cu^2+^ ions present in the silica network with danofloxacin, in the same way as occurs with other fluoroquinolones [72].

### 3.2. Ions Release Assay

Results of sample ion release are depicted in Figure 3. As shown below, the release profile of Ca is typical of mesoporous bioactive glasses with two differentiate stages: a quick release of ion products and followed by a release–precipitation equilibrium. The three samples released similar amounts of Ca^2+^ ions independently of the percentage of Cu content, reaching values of 72.7 ± 2 mg/L at 6 d. Maeno et al. [73] showed that concentrations exceeding 10 mM (40,078 mg/L) of calcium were cytotoxic to osteoblasts. Those results indicate Ca^2+^ release from the three HBGN belong to the cytocompatible range. As can be seen in the Cu-doped compositions, the release of Cu^2+^ ions continued even after 7 d, but the amount of Cu^2+^ released decreased after the third day, reaching a cumulative amount maximum of 31.6 ± 2 and 9.8 ± 0.1 mg/L for the 5 and 2.5% Cu-HBGN compositions, respectively. Both amounts are below the cytotoxicity threshold, 152 mg/L [74]. The ion release for the three compositions are promising when reaching active concentrations without becoming cytotoxic.

### 3.3. Danofloxacin Release Assay

The cumulative release of danofloxacin from each composition and the parameters of the danofloxacin release kinetics from HBGN samples are collected in Figure 4. Antibiotic release kinetics from HBGN were evaluated according to first-order kinetics model (1) with an empirical non-ideality factor (*δ*) [75].
(1)$$\frac{W_{t}}{W_{0}}={\left(\frac{W_{t}}{W_{0}}\right)}_{\max}{\left(1-e^{k_{1}t}\right)}^{\delta }$$
where *W_t_*/*W*_0_ is the percentage of antibiotic released at time *t* and *k*_1_, the release constant. *δ* values are 1 for first-order kinetic material. *δ* gives an approximation of the degree of fidelity of the proposed model to the theoretical first-order kinetic. The release profile of danofloxacin from the 0% Cu-HBGN composition was characterized by a rapid release in the first 8 h. After 24 h, a slower and sustained release is observed until 3 d, at which time the amount of danofloxacin released decreased considerably, reaching a maximum value of 16 µg/mL. Figure 4 shows that this release represented 77% of the total amount of danofloxacin loaded, which means that another 23% remained, hypothetically, inside the core of the nanoparticles. These data corroborate those obtained by Li [21], in which a two-step release is described: a rapid and abrupt release from the radial pore of the shell, and a slow and sustained release from the core.

Regarding Cu-doped samples, that samples exhibited a slower release but without depleting the amount of copper released, in fact, the trend indicates that the release continued after 7 d. As expected from their textural properties, there were no significant differences in the release obtained between the 2.5 and 5% Cu-HBGN compositions, reaching maximum values of 3 and 5 µg/mL. However, an upward trend was observed for the release of danofloxacin obtained by the 5% Cu-HBGN composition. A remarkable aspect was that the copper-doped compositions released 12 and 20% of the total danofloxacin loaded, which means that the Cu incorporation was affecting the release of danofloxacin. That interaction is taking place between Cu^2+^ cations with danofloxacin, in the same way that it occurs between said cations with different fluoroquinolones, that is, antibiotics of the same family as danofloxacin [72]. All compositions released enough amounts of antibiotic to exceed the MIC of 0.03 and 0.12 μg/mL for *E. coli* and *S. aureus*, respectively [47], in less than 8 h. However, the concentrations obtained above the MIC were maintained until 25 h for 0% Cu-HBGN, 50 h for 2.5% Cu-HBGN and 72 h for 5% Cu-HBGN. This interaction is far from being a problem, it guarantees that a burst release does not occur as if it is occurring with 0% Cu-HBGN. Instead, it is more advantageous to obtain a sustained release and sufficiently concentrated to overcome the MIC for a longer time, managing to eliminate the entire bacterial population and thus avoid the appearance of resistant strains.

### 3.4. Cells Assays

#### 3.4.1. Cytotoxicity Assays in Pre-Osteoblastic Mammalian Cells

Once the samples were characterized, cell assays were carried out to study whether these materials could affect the cellular proliferation process of the pre-osteoblastic line MC3T3-E1 cells. Cell proliferation is a biological parameter widely used to test the cytocompatibility of HBGN since it reflects the growth capacity of cells in the presence of such nanomaterials. In order to study which was the maximum concentration tolerable by the cells, HBGN samples without antibiotic loading were tested at 200 and 300 µg/mL loading.

The proliferation of pre-osteoblasts 24 h after exposure to 200 and 300 µg/mL of all compositions was similar to that obtained by cells cultured in the absence of HBGN (Figure 5). However, this proliferation significantly decreased after exposure to all the copper-enriched compositions over 72 h, the reduction being greater in pre-osteoblasts cultured with the higher concentration of Cu-HBGN samples. These results indicate that the decreased proliferation is more closely related to the total amount of copper than to the HBGN per se, as the proliferation values of pre-osteoblasts exposed to 200 µg/mL of the 5% Cu-HBGN sample were similar to those obtained when exposed to 300 µg/mL of the 2.5% Cu-HBGN sample. These values indicate that the synthesized HBGN are cytocompatible and useful for our proposal.

#### 3.4.2. Internalization of HBGN and Morphology Results in MC3T3-E1 Cells

The use of HBGN for different biomedical applications such as bone infection treatment, among others, requires that such nanomaterials have to be incorporated by the cells close to the infection without provoking structural and functional cell alterations. Once the maximum concentration of HBGN tolerable by this cell line was known, the morphology cellular and the intracellular localization of the HBGN was evaluated by fluorescence microscopy. As seen in Figure 6, the cells showed a natural phenotype, also known as Wild type, in the presence of HBGN. The images also show that these HBGN are preferentially localized in the perinuclear area as previously observed in literature [76,77], clearly outside of the nucleus and no apparent signs of nuclear shrinkage (pyknosis) was observed.

Similar images were obtained for all compositions tested. Various studies have postulated that it is an endocytosis process that can have three outcomes: nanoparticles are internalized by endocytic vesicles and a large part of them are excreted into the extracellular environment; a fraction of these endocytic vesicles travel to the cytosol where they fuse with a secondary endosome and this in turn ends up fusing with the lysosome to digest the nanoparticles; and a small number of nanoparticles escape from the endolysosome to remain in the cytosol and release the charged molecules in the nanoparticles as well as the ions that are the product of their degradation [78].

#### 3.4.3. Flow Cytometry Studies

With the aim of investigating the response of MC3T3-E1 cells to continuous exposure to HBGN for 3 d, the cell uptake process was evaluated through the intracellular complexity parameter, as well as the viability, and the ROS intracellular production by flow cytometry.

In this study, we quantitatively evaluated the incorporation of HBGN by MC3T3-E1 pre-osteoblasts by means of flow cytometry. This method is widely used to evaluate the incorporation of nanomaterials by mammalian cells through the evaluation of changes in the flow cytometric side-scatter intensity. Figure 7A shows the internal complexity of the pre-osteoblasts after exposure to the HBGN samples for 3 d. In flow cytometry, light scattering at a 90° dispersion angle is called side-scatter light (SSC), determined in part by the cellular cytoplasm, mitochondria, and pinocytic vesicles [79], phagocytosis of foreign bodies as well as the internalization of nanoparticles [80,81], its intensity being proportional to the intracellular complexity. The results revealed significant differences in the SSC parameters of pre-osteoblasts when exposed to the different HBGN samples with respect to those shown by control cells. However, no significant differences were observed between the different compositions. These results confirm the internalization of all HBGN compositions, and it is concluded that the internalization of the HBGN is independent of the atomic percentage of Cu doped in the material, as had already been observed in the internalization of the FTIC-labelled HBGN (Figure 6). In view of the results, it can be hypothesized that the following results obtained are mainly due to the degradation of the HBGN inside cells. Cell viability is an important biological parameter related to the integrity of the cell membrane, which allows the evaluation of the biocompatibility of the nanomaterials. In fact, the interaction of HBGN with cell membranes has been reported as one of the major causes of their cytotoxicity. Figure 7B shows the viability of the cells and as can be seen, there are no significant differences in the cell viability of pre-osteoblast after exposure to the different HBGN samples with respect to the control cells or between the different HBGN, so that the exposure of HBGN to a concentration of 50 µg/mL for 3 d does not affect viability in the MC3T3-E1 cell line. These results do not conflict with those obtained using MTS, since they served to establish the limits of the concentration to which this cell line could be exposed without leading to a drastic decrease in proliferation, while this test aims to see the effect of long-term viability without removing the HBGN from the well at a lower concentration.

A cell biomarker related to cell cytotoxicity caused by nanomaterials involves mostly free-radical mechanisms, the most important one being oxidative stress mediated by ROS production. Overproduction of ROS can induce an imbalance between the excessive generation of ROS and the limited antioxidant defense capacity of cells leading to adverse biological effects such as membrane lipid peroxidation, protein denaturation, mitochondrial dysfunction and DNA changes. Figure 7C shows the percentage of the population of pre-osteoblasts with ROS intracellular content. After exposure of this cell type to the different HBGN, no significant differences were obtained with respect to control cells or between the different HBGN samples. These results indicate that the cell uptake of these HBGN nanomaterials does not affect the ROS population percentage. Figure 7D shows the intracellular content of ROS (intensity, arbitrary units (A.U.)) of pre-osteoblasts after exposure to the different samples. Thus, a significant increase in the ROS intensity can be observed in the cells treated with HBGN, the highest being that of the sample with 5% Cu-HBGN (more than double) that obtained by the other HBGN compositions. However, the intensity between compositions 0 and 2.5% Cu-HBGN did not reveal significant differences. Since the only difference between the compositions is the atomic percentage of copper in HBGN, we can attribute the increase observed to the effect that Cu^2+^ cations have inside the cell. Despite these intense DCFH values, a decrease in cell viability was not obtained (Figure 7B), which may suggest that the concentrations of HBGN tested in this study are not cytotoxic. These ROS values do not necessarily correspond to a cytotoxic response to the material, most cell lines maintain a basal expression as a consequence of the mitochondrial electron transport chain necessary to catabolize carbohydrates, fatty acids and amino acids [82]. These ROS values increased during the proliferation process as a consequence of the bioenergetic requirement necessary for cell cycle progression and entry into mitosis. [83]. In view of these results, it can be hypothesized that copper release (Figure 3) at this concentration does not trigger a cytotoxic response but is effective in promoting proliferation as observed in our results (Figure 7).

### 3.5. Bacterial Assays

#### 3.5.1. Bacterial Growth Inhibition Tests in the Planktonic State

HBGN loaded and unloaded with danofloxacin were tested at 200, 100 and 10 µg/mL for *E. coli* and 260, 130 and 20 µg/mL for *S. aureus*. No decrease in bacterial proliferation was obtained in samples without antibiotic loading for *E. coli* or *S. aureus* at the concentrations used (data not shown). These results are to be expected given that the literature reports effective Cu^2+^ concentrations of around 230 and 600 mg/L for *E. coli* and *S. aureus*, respectively [84]. However, the authors have observed the efficacy of the bactericidal capacity of copper when concentrations of 2 mg/mL of nanoparticles Cu doped are used [63].

As can be seen in Figure 8A, the nanosystem was shown to be an effective strategy, by inhibiting bacterial growth by 100% against *E. coli* at concentrations equal to or greater than 10 μg/mL. Figure 8B shows the results obtained against *S. aureus*. It was observed that a concentration of 20 µg/mL was sufficient to obtain a 99.9% reduction in bacterial growth with respect to the control, for all HBGN compositions loaded with danofloxacin. No significant differences were obtained between the different compositions. These results allow us to conclude that the MBC with this nanosystem is 10 µg/mL for *E. coli* and 20 µg/mL for *S. aureus*, concentrations much lower than those used by other similar nanosystems described in the literature [61,63].

#### 3.5.2. Halo Inhibition Assays

Assays were carried out using the Kirby–Bauer methodology, testing 300, 200 and 100 μg/mL of each composition loaded or not-loaded with danofloxacin using *E. coli* and *S. aureus* as bacteria models.

Compositions not loaded with danofloxacin did not produce any growth inhibition halo for *E. coli* or *S. aureus* (results not shown). As shown in Figure 9A, halos were observed at 6 h for each HBGN composition loaded with danofloxacin at all three compositions used. Similar results were obtained at 24 h of exposure to each HBGN compositions (Figure 9B). No significant differences were observed between the three samples at each concentration used at 6 h or 24 h. Figure 9C,D shows the efficacy of the nanosystem loaded with danofloxacin against *S. aureus*. The 2.5% Cu-HBGN at 300 micrograms and 5% Cu-HBGN at 200 micrograms were effective in inhibiting the growth of *E. coli*.

These results indicate that for this type of assay the copper activity is not sufficient to produce any inhibition halo for *E. coli* or for *S. aureus*. However, against *S. aureus*, a possible synergistic effect between the copper ions and danofloxacin could be observed, as inhibition halos were obtained only in the Cu-doped compositions at 300 μg/mL for the 2.5% Cu-HBGN composition and 200 and 300 μg/mL for 5% Cu-HBGN. The difference in the efficacy of the nanosystem in this test compared to the bacterial growth inhibition test in the planktonic state is explained below.

Bacteria in a sessile state are faced with a more aggressive environment with limited moisture and nutrients, which is why they form aggregates and change their physiology, becoming metabolically slower but more resistant. However, bacteria in a planktonic state have a quantity of richer and more accessible nutrients drastically increasing metabolism and motility, thus exposing them more to the effect of antibiotics [85,86,87], so it is common that the concentrations of antibiotics needed to kill the same bacteria in a sessile state compared to planktonic bacteria are typically in the order of a thousand times higher [88].

It is remarkable that, although the 2.5 and 5% Cu-HBGN compositions released less danofloxacin than 0% Cu-HBGN, they were more effective in inhibiting bacterial growth in *E. coli* and *S. aureus*. In fact, this strategy is more effective than using high concentrations of Cu^2+^ [84], as observed in microbiological assays where neither a decrease in viability in the planktonic state nor halos in the sessile state were obtained, and high concentrations of antibiotic, as in the case of 0% Cu-HBGN, which released higher amounts of danofloxacin, and was even more efficient than other systems where larger quantities of material are needed [62,63,66,74].

#### 3.5.3. Biofilm Degradation Assay

Finally, to verify the ability of the nanosystem to degrade a biofilm, the preformed biofilm of *S. aureus* was used as a model due to its greater resistance than that formed by *E. coli*. For this test, 200 μg/mL of each composition loaded with danofloxacin was used at the maximum cytocompatible concentration. The results of this test are shown in Figure 10.

The control (Figure 10A) showed a dense biofilm layer throughout the field with a thickness of approximately 70 nm, no bacterial death was observed. The Cu-free HBGN loaded (Figure 10B), reduced the thickness of the biofilm by approximately 16% ± 8 with respect to the control, and a darkening of the image could be observed due to the appearance of red pixels corresponding to bacterial death. The exopolysaccharide matrix could not yet be seen in the image due to the huge bacterial population that remained alive after exposure to this composition. Figure 10C,D show the effects of treatment with 2.5% and 5% Cu-HBGN loaded with danofloxacin. There is clear evidence of a reduction in biofilm thickness of approximately 52 ± 18% with respect to the control, obtained with both compositions. For the sample 2.5% Cu-HBGN (Figure 10C), an increase in bacterial death and loss of extracellular matrix was observed, marked in red and blue respectively. Finally, Figure 10D shows the biofilm treated with 5% Cu-HBGN with danofloxacin. Although the biofilm reduction obtained by the 5% Cu-HBGN is similar to that obtained by the 2.5% Cu-HBGN composition, fields of high bacterial death are observed, as well as total loss of biofilm in certain regions of the field.

These results indicate, once again, that compositions enriched with a greater amount of copper trigger greater bacterial death and greater degradation of the biofilm pre-formed by *S. aureus,* depending on the amount of copper released.

As already observed in the danofloxacin release study (Figure 4), the Cu-free composition released more danofloxacin and in less time, while the Cu-doped HBGN released less and more slowly the danofloxacin contained inside. This shows that the degradation of the biofilm is not due to a release of danofloxacin at higher concentrations than the other compositions. However, it does suggest that the lethality of the nanosystem is dependent on the amount of copper being released into the surrounding medium, as was also observed in the growth inhibition and halo inhibition assays (Figure 8 and Figure 9).

The joint action of the antibiotic and copper ion release is able to overcome the individual action of the copper ions and the antibiotic at concentrations well above those required by the joint nanosystem, possibly due to the formation of coordination complexes in which the antibiotic activity is enhanced as suggested in literature [89].

The results obtained from the microbiological assays meet several objectives at the same time since they are not cytotoxic with mammalian cells (Figure 5, Figure 6 and Figure 7) while they are effective against the inhibition of bacterial growth and the degradation of a biofilm (Figure 8, Figure 9 and Figure 10), at the same or lower concentration of HBGN samples, in two of the bacterial strains which cause the majority of bone infections.

Moreover, this work opens up the possibility of using functionalizing agents that provide further applications by anchoring functionally active groups to the reactive surface of the material. In this respect, it should be explored whether the nanosystem can be further improved, or whether even more effective interactions between copper ions and antibiotics of the danofloxacin family can be produced. Further research is needed to demonstrate the full potential of this nanosystem for the treatment of bone infections.

## 4. Conclusions

A nanosystem based on hollow bioactive glass nanoparticles in the SiO_2_-P_2_O_5_-CaO-CuO system loaded with danofloxacin was synthesized and characterized to be used for the local treatment of bone infections.

The three compositions investigated containing 0, 2.5 or 5% of CuO showed high values of specific surface area and pore volume and two different sizes of pores which made them versatile for loading with diverse biomolecules and drugs.The nanosystem was able to release biologically effective amounts of therapeutic inorganic ions and danofloxacin. The Cu-loading of the HBGN produced more gradual release of danofloxacin that was extended for over one week.Assays with MC3T3-E1 pre-osteoblasts showed a biocompatible behavior after 3 d of contact of the nanoparticles with cells.Microbiological assays showed that copper ions enhanced the bactericidal effect of danofloxacin against *E. coli* and *S. aureus* in both planktonic and sessile state. Moreover, the nanosystem was able to degrade a biofilm preformed of *S. aureus* at minimal concentration, suggesting its suitability as a bactericide agent.

The results shown in this study demonstrate that the amount of antibiotic danofloxacin and copper ions needed to eliminate a bacterial population is drastically reduced when a nanosystem that effectively combines both strategies is employed. Consequently, we can conclude that HBGN-based nanosystems are promising alternatives in the fight against bone infections.

## Data Availability

Not applicable.

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
