# Peer review of "Cu-Doped Hollow Bioactive Glass Nanoparticles for Bone Infection Treatment"

_pharmaceutics, 2022, doi:10.3390/pharmaceutics14040845_

Round 1

Reviewer 1 Report

In the submitted manuscript entitled „Cu-doped hollow bioactive glass nanoparticles for bone infection treatment“a new approach for bone infections, based on Cu-doped hollow bioactive glass, was developed. The proposed research is very interesting and is in the field of biomaterials that is highly studied in recent years. I will give comments on the submitted manuscript, and I hope my suggestions will help authors to increase the quality of the manuscript.

Abstract:
The abstract is very well written, easy to read and nicely summarizes what is done in the manuscript.

Introduction:
The introduction is nicely written and includes all the needed information to understand why this work is important. There is no need for modification of the written text.

Materials and Methods
The part „(All reagents were purchased from Sigma-Aldrich, St. Louis, MO, USA)“ should be without brackets. It would be better as it is a separate sentence.
The quality of Scheme 1. should be increased.
On page 4, line 164, the dot (.) at the end of the sentence is missing.
Bracket „(CellTiter 96® AQueous Assay 185 (Promega, Madison, WI, USA))“ should be written as  (CellTiter 96® AQueous Assay 185, Promega, Madison, WI, USA)“.
The methodology design is appropriate for obtained biomaterials. Overall, this section is written properly. All experiments could be replicated by other researchers according to the written protocols.

Results and Discussion
EDS Spectra should be shown below TEM micrographs, with the size same as micrographs. This suggestion is because the EDS results can not be properly seen in Figure 1. In addition, the circled yellow square in Figure 1. should be bigger as they show crucial information.
Page 8, lines 311-312. Can the authors explain why the incorporation of phosphorus was low?
Authors should compare their results with results obtained in other studies (for other ions) along with copper. In addition, the authors should discuss the antibacterial mechanisms of these ions on the bacteria cell wall.
Overall, authors should work on the quality and presentation of images.

Overall, the quality of the submitted manuscript is at a high level. I try to find drawbacks; however, there were only a few that I mentioned in my comments. After the authors accept my suggestions, I suggest that the manuscript is accepted for publication.

Author Response

Dear Reviewer 1

We thank you very much for the effort you have made in reviewing our manuscript. All the comments and suggestions were taken into consideration and we have done our best to answer all questions and make all necessary modifications in the manuscript. To facilitate observation of the corrections, they were underlined in yellow in the revised version of the manuscript. The reviewer' comments and the authors' responses follow:

1- The part „(All reagents were purchased from Sigma-Aldrich, St. Louis, MO, USA)“ should be without brackets. It would be better as it is a separate sentence.

Following the referee suggestion, the brackets were eliminated.

2-The quality of Scheme 1. should be increased.

Authors appreciate your comments. Scheme 1 has been redrawn for better visualization.

3- On page 4, line 164, the dot (.) at the end of the sentence is missing.

The dot was added.

4-Bracket „(CellTiter 96® AQueous Assay 185 (Promega, Madison, WI, USA))“ should be written as  (CellTiter 96® AQueous Assay 185, Promega, Madison, WI, USA)“.

The bracket was eliminated

5- EDS Spectra should be shown below TEM micrographs, with the size same as micrographs. This suggestion is because the EDS results can not be properly seen in Figure 1. In addition, the circled yellow square in Figure 1. should be bigger as they show crucial information.

The authors are grateful for their comments in order to facilitate the understanding of this work. The size of the components in Figure 1 has been increased.

6- Page 8, lines 311-312. Can the authors explain why the incorporation of phosphorus was low?
Following the reviewer's recommendations, the following paragraph has been added: “Phosphorus integration is usually reduced due to the slow hydrolysis rate of the phosphorus precursor TEP [47]. Despite the low incorporation rate observed also in other works [48-49], it was decided to keep the addition of TEP as it allows a higher incorporation of calcium into the nanoparticles [49].

7- Authors should compare their results with results obtained in other studies (for other ions) along with copper. In addition, the authors should discuss the antibacterial mechanisms of these ions on the bacteria cell wall.

Following the reviewer's recommendations, we have added bibliography, distributed throughout the results and discussion section, which supports the superior textural and bactericidal properties of our nanosystem with respect to the one currently described in the literature.

In addition, a paragraph has been introduced in the introduction explaining the mechanisms of action of Cu2+ ions described in the literature with bactericidal effects: “In addition, the bactericidal activity of copper against E. coli, B. subtilis and S. aureus has been described in the literature [40,41]. Although the mechanisms of action are still under discussion, it has been reported that Cu2+ ions bind to the bacterial cell wall by electrostatic attraction, due to the fact that the bacterial wall is densely populated by negatively charged peptidoglycan molecules, the attraction being greater for Gram-negative bacteria [42]. This leads to solidification of the structural proteins that compose it, irreversibly damaging it [43,44]. The damage may be mediated by the production of reactive hydroxyl radicals that are known to be involved in numerous reactions detrimental to proper cell development, such as oxidation of proteins and lipids [45]”

Reviewer 2 Report

In this manuscript, the authors introduced a new approaches to treat bone infection and prevent drug resistance, which is a nanosystem based on hollow bioactive glass nanoparticles in the SiO2–P2O5–CaO–CuO system loaded with danofloxacin. It provides us a new strategy for the local treatment of bone infection. I would like to put forward the following questions and comments:

  1. In Figure 1, the same scale bar was used in three TEM pictures, but in the middle one, the size of the nanoparticles look smaller than those in other two pictures. Some explanation or correction is needed.
  2. In line 240, there is a mistake about Staphylococcus aureus, it should be Gram-positive.
  3. There is no in vivo study in this manuscript, it would be more convincing if in vivo study could be added.

Author Response

Dear Reviewer 2:
We thank you very much for the effort you have made in reviewing our manuscript. All the comments and suggestions were taken into consideration and we have done our best to answer all questions and make all necessary modifications in the manuscript. To facilitate observation of the corrections, they were underlined in yellow in the revised version of the manuscript. The reviewer' comments and the authors' responses follow:
1 - In Figure 1, the same scale bar was used in three TEM pictures, but in the middle one, the size of the nanoparticles looks smaller than those in other two pictures. Some explanation or correction is needed
The authors are grateful for their comments. The synthesis methodology described in this article allows obtaining nanoparticles with a slightly variable size. As indicated on line 298, the size of the nanoparticles is about 190 ± 65 nm. In order to observe the nanoparticles of the 2.5% Cu-HBGN composition, the microscopy technician had to zoom in and for this reason the dimension is slightly larger.

2- In line 240, there is a mistake about Staphylococcus aureus, it should be Gram-positive.

Authors appreciate your review. The typo has been checked.

3- There is no in vivo study in this manuscript, it would be more convincing if in vivo study could be added.

Thank you for your comment. The in vivo assays are scheduled for further work, due to lack of time, they could not be carried out yet.
